# Adipose and Plasma microRNAs miR-221 and 222 Associate with Obesity, Insulin Resistance, and New Onset Diabetes after Peritoneal Dialysis

**DOI:** 10.3390/nu14224889

**Published:** 2022-11-18

**Authors:** Gordon Chun Kau Chan, Win Hlaing Than, Bonnie Ching Ha Kwan, Ka Bik Lai, Ronald Cheong Kin Chan, Jeremy Yuen Chun Teoh, Jack Kit Chung Ng, Kai Ming Chow, Phyllis Mei Shan Cheng, Man Ching Law, Chi Bon Leung, Philip Kam Tao Li, Cheuk Chun Szeto

**Affiliations:** 1Carol & Richard Yu Peritoneal Dialysis Research Centre, Department of Medicine & Therapeutics, The Chinese University of Hong Kong, Hong Kong 999077, China; 2Li Ka Shing Institute of Health Sciences (LiHS), Faculty of Medicine, The Chinese University of Hong Kong, Hong Kong 999077, China; 3Department of Anatomical & Cellular Pathology, The Chinese University of Hong Kong, Hong Kong 999077, China; 4S.H. Ho Urology Centre, Department of Surgery, The Chinese University of Hong Kong, Hong Kong 999077, China

**Keywords:** adipose miRNA, peritoneal dialysis, obesity, insulin resistance, diabetes mellitus

## Abstract

Background: The correlation between microRNA, obesity, and glycemic intolerance in patients on peritoneal dialysis (PD) is unknown. We aimed to measure the adipose and plasma miR-221 and -222 levels, and to evaluate their association with adiposity, glucose intolerance, and new onset diabetes mellitus (NODM) after the commencement of PD. Methods: We prospectively recruited incident adult PD patients. miR-221 and -222 were measured from adipose tissue and plasma obtained during PD catheter insertion. These patients were followed for 24 months, and the outcomes were changes in adiposity, insulin resistance, and NODM after PD. Results: One hundred and sixty-five patients were recruited. Patients with pre-existing DM had higher adipose miR-221 (1.1 ± 1.2 vs. 0.7 ± 0.9-fold, *p* = 0.02) and -222 (1.9 ± 2.0 vs. 1.2 ± 1.3-fold, *p* = 0.01). High adipose miR-221 and -222 levels were associated with a greater increase in waist circumference (miR-221: beta 1.82, 95% CI 0.57–3.07, *p* = 0.005; miR-222: beta 1.35, 95% CI 0.08–2.63, *p* = 0.038), Homeostatic Model Assessment for Insulin Resistance (HOMA) index (miR-221: beta 8.16, 95% CI 2.80–13.53, *p* = 0.003; miR-222: beta 6.59, 95% CI 1.13–12.05, *p* = 0.018), and insulin requirements (miR-221: beta 0.05, 95% CI 0.006–0.09, *p* = 0.02; miR-222: beta 0.06, 95% CI 0.02–0.11, *p* = 0.002) after PD. The plasma miR-222 level predicted the onset of NODM (OR 8.25, 95% CI 1.35–50.5, *p* = 0.02). Conclusion: miR-221 and -222 are associated with the progression of obesity, insulin resistance, and NODM after PD.

## 1. Introduction

Peritoneal dialysis (PD) is an effective home-based dialysis therapy for patients with advanced chronic kidney disease (CKD). The alleviation of uremic-related anorexia and additional calorie intake from glucose-containing dialysate [1] often cause obesity in patients on PD, especially during the early initiation period [2,3,4,5]. The additional glucose absorption can also induce hyperglycemia, insulin resistance, and new-onset diabetes mellitus (NODM) after the commencement of PD [6]. Patients who developed these features are prone to adverse outcomes such as a rapid decline in residual renal function, cardiovascular events, PD technique failure that requires conversion to haemodialysis, and mortality [4,7,8,9].

MicroRNAs (miRNA) are short, non-coding RNA molecules that carry crucial regulatory functions [10,11,12]. These miRNAs, which are produced in adipose tissue, modulate post-transcriptional gene expression, cellular proliferation, differentiation, and death. The role of miRNAs, namely miR-221 and -222, in adipogenesis, insulin resistance, and metabolic diseases [13,14,15,16,17] has been increasingly recognized. However, their relationship with dialysis-associated obesity and glucose intolerance has not been adequately explored. While the identification and quantification of miRNAs may assist clinicians to detect high-risk patients [18], these miRNAs also represent a potential target for future miRNA-based therapy to improve patient outcomes.

In this study, we measured the miR-221 and -222 levels in the adipose tissue and plasma samples obtained from a cohort of incident PD patients. We also evaluated their association with obesity, insulin resistance, and NODM after the commencement of PD.

## 2. Materials and Methods

### 2.1. Study Design

This was a single-center prospective cohort study, which was approved by the Joint Chinese University of Hong Kong—New Territories East Cluster Clinical Research Ethics Committee (Reference Number CREC-2008.554). All study procedures were compliant with the Declaration of Helsinki. Consecutive incident PD patients were recruited from 1 January 2011 to 31 December 2013 at the Prince of Wales Hospital. Patients with active malignancy, or on anabolic steroid treatment for at least 4 weeks, were excluded. Written informed consent was obtained at study enrolment. After consent, a blood sample was obtained on the day of PD catheter insertion, and was processed, centrifuged, and stored at −4 °C overnight. During the PD catheter insertion by mini-laparotomy, 1–2 g of subcutaneous and pre-peritoneal adipose tissue sample was obtained, processed immediately and stored at −80 °C overnight.

### 2.2. Extraction of RNA and miRNA

The extraction of RNA and miRNA was described in detail previously [10,11]. In essence, RNA was extracted by the miRNeasy minikit from Qiagen following the principle of combined phenol/guanidine-based lysis of samples with silica membrane-based purification of total RNA with Spin column format. QIAzol Lysis Reagent, included in the kits, is a monophasic solution of phenol and guanidine thiocyanate, designed to facilitate lysis of adipose tissues, to inhibit RNases and to remove most of the cellular DNA and proteins from the lysate by organic extraction. As for plasma RNA extraction, the relative quantification of gene expression was first processed with the following materials of each step with their specific manufacturers’ instructions, followed by the total RNA extraction by Qiagen miReasy serum/plasma advanced kit (217204) (Hilden, Germany). The extracted total RNA was kept at −80 °C until downstream experiments.

The cDNA template was then prepared by reverse transcription with Taqman Advanced miRNA cDNA synthesis kit (Thermo Fisher Scientific, Waltham, MA USA) (A28007), a universal kit covering downstream targets miR-221 and -222. miR-103a was used as a housekeeping gene for quality control because of its stability [19,20]. Amplification of these targets was achieved by Applied Biosystems Step One plus RT-PCR (Thermo Fisher Scientific, Waltham, MA USA) through specific Taqman Advanced miRNA assays in the presence of Taqman Fast Advanced Master Mix. (Thermo Fisher Scientific, Waltham, MA USA) Names (Assay ID) of specific Taqman Advanced miRNA essays were hsa-miR-221-3p (477981_mir) for miR 221, and hsa-miR-222-3p (477982_mir) for miR-222. In short, cDNA was diluted 1:10 and then 5 µL of diluted cDNA was added to PCR mix of 10 µL of TaqMan^®^ Fast Advanced Master Mix (2×), 1 µL of specific Taqman Advanced miRNA essays, and 4 µL of RNase free water (total 20 µL per reaction in each patient’s target). Triplicates were performed for qPCR for optimization. Step One software version 2.3 (Thermo Fisher Scientific, Waltham, MA USA) was used for detection of amplification and relative quantification (∆∆Ct method) was applied for expression of targets in fold compared with the levels detected in samples from six healthy subjects [21].

### 2.3. Anthropometric Measurements

Body weight (BW) and height (BH) were measured. Body mass index (BMI), calculated by body weight (in kg) divided by the square of body height (in m), was used to assess general obesity. Central obesity was evaluated by waist circumference. All anthropometric measurements were performed annually with an emptied abdomen (i.e., after drainage of dialysate).

### 2.4. Glycemic Profile

Fasting plasma glucose, glycated haemoglobin A1c (HbA1c), insulin, and C-peptide levels were measured. Insulin resistance was evaluated by the Homeostatic Model Assessment (HOMA) index, which is calculated by: fasting plasma insulin (mU/L) × fasting plasma glucose (mmol/L)/22.5. These parameters were repeated annually. We also recorded the daily insulin requirements at a 6-month interval.

### 2.5. Other Clinical and Biochemical Parameters

Baseline clinical and laboratory data were obtained by chart review. Clinical data comprise patient’s age, gender, primary diagnosis of renal disease, concomitant chronic medical illnesses including diabetes mellitus, ischemic heart disease, cerebrovascular accident, and peripheral vascular disease. Residual renal function was measured by the residual glomerular filtration rate from 24 h collection of urine. The Charlson Comorbidity Index (CCI) was used to assess the comorbidity load. Nutrition was measured by serum albumin and normalized protein catabolic rate (nPCR). Peritoneal characteristics were determined by a standard peritoneal equilibrium test as described by Twardowski. Dialysate-to-plasma ratios of creatinine at 4 h (D/P4) and mass transfer area coefficients of creatinine (MTAC) normalized for body surface area were calculated. Concurrent medication use, which may potentially induce DM and obesity, namely statins, beta-blockers, thiazide diuretics, antiepileptics, antidepressants, and antipsychotics, was also analyzed.

### 2.6. Follow-Up and Outcome Measurements

All patients were followed for 24 months. After recruitment, all of them received standard dietary counseling by experienced dietitians at the time of dialysis training. In general, they were advised to have energy intake of 25 to 35 kcal/kg/day (depending on their body mass index (BMI)), 25–30% of which was fat, and protein, 1.0–1.2 g/kg/day. Clinical management and dialysis regimens were decided by attending nephrologists and were not affected by the study. Renal transplantation, conversion to haemodialysis, and transferal to other dialysis units were defined as censoring events. The primary outcome was the change in WC. Secondary outcomes were changes in BMI, HOMA index, and insulin requirements. We also determined the incidence of NODM among non-DM CKD patients, which was diagnosed according to the American Diabetes Association guidelines by fasting plasma glucose greater than 7 mmol/L, or HbA1c greater than 6.5 on two occasions [22].

### 2.7. Statistical Analysis

Statistical analysis was performed by SPSS for Mac software version 27 (SPSS Inc., Chicago, IL, USA). Descriptive data were presented as mean ± SD if normally distributed, or median (interquartile range) otherwise. Clinical parameters were compared by Student’s *t*-test, chi-square test and one-way analysis of variance as appropriate. Correlation was compared by Spearman’s rank correlation. The miRNA expression was divided according to tertiles for comparison. Multivariate linear regression model was then constructed to identify significant predictors of outcomes after adjusting for confounders. Factors that were potential contributors to outcome, such as age, pre-existing DM, high-sensitive C-reactive protein, D/P4, daily dextrose load, baseline HOMA, nPCR, BMI, and WC were included in the model and were listed in the corresponding tables. Kaplan–Meier plots were constructed, and log-rank test was used to compare the rate of NODM between the curves of miRNA levels dichotomised according to median. Multivariate Cox proportional hazards models were further constructed to identify independent predictors. The factors included in the models were age, gender, fasting plasma glucose, high-sensitive C-reactive protein, daily dextrose load, prescription of icodextrin-containing dialysate, BMI, and WC. *p* < 0.05 was defined as statistically significant. All probability tests were two-tailed.

## 3. Results

### 3.1. Clinical Characteristics

We recruited 165 incident PD patients. Their clinical characteristics are summarized in Table 1. Their average age was 58.4 ± 11.1 and 126 (76.4%) of them were male. Among them, 85 (51.5%) patients had diabetic nephropathy, and 104 (63.0%) patients had pre-existing DM. The average adipose miR-221 and -222 levels in the PD cohort, compared to the healthy individuals, were 10.6 ± 1.0-fold and 38.1 ± 1.8-fold, respectively. The average plasma miR-221 and -222 levels were 6.4 ± 2.8-fold and 0.8 ± 0.1-fold, respectively. miR-221 and -222 levels correlated significantly at adipose (r = 0.92, *p* < 0.001) and plasma (r = 0.64, *p* < 0.001) levels. However, there was a lack of correlation between adipose and plasma miR-221 (*p* = 0.7), and between adipose and plasma miR-222 (*p* = 0.3).

Patients with pre-existing DM had significantly higher adipose miR-221 (1.13 ± 1.20 vs. 0.70 ± 0.85-fold, *p* = 0.016) and -222 (1.90 ± 1.97 vs. 1.16 ± 1.29-fold, *p* = 0.01; Table 2). Plasma and adipose miR-221 and -222 levels also correlated significantly with baseline glycaemic parameters, such as fasting plasma glucose, glycated HbA1c, insulin, HOMA, and serum albumin as shown in Appendix A. However, miR-221 and -222 did not correlate with baseline BMI and WC. 

### 3.2. Changes in Adiposity

After 299.1 patient years, 27 patients died, 7 patients received kidney transplantation, 10 patients were converted to haemodialysis and 1 patient was transferred to another dialysis center (Figure 1). One hundred and twenty patients survived and remained on PD in the 24th month. Their anthropometric parameters are summarized in Table 3. During the first 12 months of PD, all anthropometric parameters such as WC, BW, and BMI increased significantly, while only WC continued to rise in the second year (Table 3). The magnitude of increase in WC (+4.48 ± 6.82 vs. +2.06 ± 5.08 cm, *p* = 0.029), and BMI (+1.19 ± 1.66 vs. +0.44 ± 2.04 kg/m^2^, *p* = 0.04) were more significant in non-DM CKD patients.

The changes in BMI and WC increased in a stepwise manner across the tertiles of adipose miR-221 and -222 levels (Figure 2 and Appendix A). The magnitude of changes was more pronounced among non-DM patients (Figure 2). However, the changes of BMI and WC did not differ across plasma miR-221 and -222 tertiles (Appendix A). After adjusting for confounders in the multivariate model, adipose miR-221 predicted BMI and WC change after PD. In the same model, adipose miR-222 also predicted WC change (Table 4). Dialysate dextrose load, serum albumin, high-sensitive C-reactive protein levels, and antidepressant use were other associated factors identified.

### 3.3. Changes in Glycaemic Parameters

Fasting plasma glucose, insulin, and HOMA increased significantly during the first 12 months of PD, which then plateaued in the next 12 months (Table 3). The daily insulin requirements also increased, though the changes were only statistically significant during the first 6 months of PD (*p* = 0.006) (Table 3). Patients with high adipose miR-221 and -222 had a significantly greater rise in fasting plasma insulin level, HOMA, and insulin requirements after PD (Figure 3). However, the changes in glycaemic parameters were not significant between plasma miR-221 and -222 tertiles (Appendix A). After adjusting for the confounding effects in the multivariate model, adipose miR-221 and -222 remained as significant predictors of a greater increase in plasma insulin, HOMA, and insulin requirements after 12 months of PD (Table 4).

### 3.4. New Onset Diabetes Mellitus after Peritoneal Dialysis

With time on PD, 26 (42.6%) patients developed NODM at an average onset time of 10.0 ± 5.7 months, with the incidence rate of 0.31 per patient year (Appendix A). The majority of them (19 patients, 73.1%) developed NODM during the first 12 months of PD. Patients who developed NODM were more general (BMI: 24.6 ± 5.4 vs. 22.2 ± 3.1 kg/m^2^, *p* = 0.04) and central obese (WC: 88.7 ± 13.5 vs. 79.9 ± 10.2 cm, *p* = 0.009) at study enrolment. A significantly higher percentage of patients with high plasma miR-222 (80.0% vs. 27.3%, log-rank test, *p* = 0.015, Figure 4) developed NODM. The numerical incidence rate of NODM was also higher for patients with high plasma miR-221, although the difference did not reach significance (66.7% vs. 33.3%, *p* = 0.2). Neither adipose miR-221 (45.5% vs. 42%, *p* = 0.9) nor adipose miR-222 (35.7% vs. 44.7%, *p* = 0.6) predicted NODM. After adjusting for confounders in the multivariate analysis, high plasma miR-222 remained as the only predictor of NODM after PD (odds ratio: 8.25, 95% CI 1.35–50.5, *p* = 0.02). Baseline WC also predicted the onset of NODM in univariate analysis (*p* = 0.03), but its significance disappeared in the multivariate model.

## 4. Discussion

In the present study, we quantified the levels of two miRNAs: miR-221 and -222, in the adipose tissue and plasma samples obtained from a cohort of incident PD patients. In short, patients with high adipose miR-221 and -222 became significantly more obese after the commencement of PD. These patients were also susceptible to glycaemic intolerance with time. In addition, non-DM CKD patients with high plasma miR-222 were prone to NODM after PD. To our knowledge, this is the first report that elucidates the association of specific miRNAs in PD-associated obesity and insulin resistance. Our result supports the roles of miR-221 and -222 in the development and progression of metabolic diseases in advanced CKD and after PD. We also provided novel supportive evidence on the use of miRNA assaying in risk stratification and prognostication.

The global uptake of PD has increased over the past decade in many parts of the world, including the USA [23]. Compared to haemodialysis, PD incurs a lower cost and less demand on infrastructure, staff expertise, and patient training [24,25]. As a home dialysis therapy, it also minimizes frequent hospital visits, which is particularly advantageous amid the novel Coronavirus-2019 (COVID-19) pandemic. With that, the burden of metabolic complications after PD, such as excessive weight gain and adiposity [5,26,27], will become global challenges to tackle among the dialysis population. The alleviation of uremic anorexia, improvement in general health, additional caloric intake from glucose-containing dialysate use, and dialysis-associated inflammation exacerbate adipogenesis [2]. Our result is in line with a past study that showed non-DM patients had more fat accumulation when undergoing PD [5]. The association of aberrant miR-221 and -222, obesity and metabolic diseases has been extensively reported in animal and in non-CKD human models [18,28,29,30,31,32,33]. These miRNAs promote adipogenesis by inhibiting the matrix metalloproteinase and nitric oxide synthesis that regulate the peroxisome proliferator-activated receptor gamma coactivator 1-alpha (PGC-1α), and nuclear factor kappa-light-chain-enhancer of activated B cell (NF-κB) signaling pathways [34]. Nevertheless, miR-221 inhibits the DNA Damage Inducible Transcript 4 (DDIT4)-mediated mammalian target of the rapamycin complex 1 (mTORC1) pathway and activates the AdipoR that alters the downstream adiponectin-related events such as lipolysis, fatty acid oxidation, and ketogenesis. Interestingly, while central and general obesity progressed in the first year, only central obesity perpetuated afterwards. Indeed, the changes in BMI can be confounded by fluid and lean tissue loss, which are common in patients on PD. Waist circumference is therefore a more reliable marker of abdominal fat load as it measures the amount of visceral and subcutaneous fat deposition in the abdominal wall, which tend to accumulate with PD [35]. The feasibility of reversing the PD-associated adipogenic process by miR-221-targeted therapy should be further explored since a pre-clinical study on an animal model demonstrated a remarkable effect against obesity with the pharmacological inhibition of miR-221 [33].

Hyperglycaemia, insulin resistance, and NODM are other common metabolic complications after PD. Our reported NODM rate of 42.6% is higher than the rate of 19–23% reported in past studies [5,36,37,38,39]. Patient ethnicity, physical activity state, dietary and cultural factors, and the utilization of glucose-sparing dialysate may account for the difference. Nevertheless, the diagnostic criteria of NODM used in these studies were inconsistent [9]. In our study, we used the latest ADA guidelines that have incorporated the additional glycated HbA1c criteria [22]. It is expected that more subjects will be classified as NODM with the new criteria. miR-221 impairs insulin sensitivity through mimicking peroxisome proliferator-activated receptor (PPAR) activation [30], upregulation of the FASN protein [30], and suppression of V-ets erythroblastosis virus E26 oncogene homolog 1 (ETS1) [30] and Sirtuin-1 (SIRT1) protein level [40]. Similarly, miR-222, which shares a homogenous gene clustering and seed sequences with miR-221 [41], reduces insulin receptor substrate (IRS)-1 gene transcription, IRS-1 substrate [42,43], estrogen receptor-alpha protein, and insulin-sensitive membrane transporter glucose transporter 4 (GLUT4) protein [13] production. miR-222 also promotes inflammation and insulin resistance through the tumor necrosis factor (TNF)-alpha-mediated pathways [28,44]. Nevertheless, miR-221 and -222 affect the production and secretion of insulin through impairing the maturation of pancreatic beta cell secretory vesicles [32], and regeneration of pancreatic beta cells [45]. Apart from the peritoneal characteristics that control the dialysate glucose absorption to systemic circulation, traditional factors such as obesity [5], chronic inflammation [46], and protein energy wasting [47] contribute to PD-related NODM [9]. However, the significance vanishes after we take the plasma miR-222 into account in the multivariate model. As discussed earlier, miR-222 induces visceral fat accumulation, which produces a broad range of pro-inflammatory cytokines that hasten glycaemic intolerance [48]. miR-222 is also involved in the innate immune response and accentuates inflammation through TNF-alpha and other associated pathways [28,44,49]. Multiple studies have supported the role of miR-222 in the pathogenesis of glucose intolerance, [16], pre-DM [16,50], gestational DM [13,51], and several inflammatory diseases [34]. Since patients with high circulating miR-222 are prone to NODM, close monitoring, regular dietitian counseling, and the avoidance of a high-glucose-content PD regime should be provided to improve their outcomes.

Intriguingly, aberrant adipose and plasma levels of the same miRNA have different metabolic effects. The association between adipose miR-221 and -222 levels and the progression of obesity and HOMA supports the paracrine function of these miRNAs at the intra-abdominal fatty tissue that we harvested. Nevertheless, adipose tissue only contributes to part of the circulating miRNAs since miRNAs are also produced in organs such as the brain, heart, liver, skeletal muscle, spleen, etc. [33,43]. While these miRNAs are preferably produced in white adipose tissue [43], a specific membrane transporter is also required for their diffusion into the bloodstream [52], which supports the lack of correlation between adipose and plasma miRNA levels in the current study. Therefore, the plasma miRNA level reflects the endocrine action with the cumulated systemic effect. In addition, our patients with DM had a numerically lower plasma miR-222, which is in contrary to the existing literature showing a higher miR-222 in the general population with type 2 DM and gestational DM [50,51]. The paradoxical observations can be explained by the distinct biochemical characteristics in diabetic patients with advanced CKD. It has been reported that nearly one third of diabetic patients with CKD [53], and 17% of diabetic patients on PD [54], have normal HbA1c even without antidiabetic agent use. The decreased renal and hepatic insulin clearance, decline in renal gluconeogenesis, deficient catecholamine release, diminished food intake, and defective energy utilization when renal function declines lead to the spontaneous improvement of glycemic control—a phenomenon called “burnt-out diabetes” [53,54]. Nonetheless, the accumulation of low-molecular-weight proteins such as RNases in CKD alter the turnover and degradation of circulating miRNAs [55]. Past study has therefore shown a different circulating miRNA profile in patients with CKD compared to those in healthy individuals [56]. As a strength of this study, we characterized and quantified adipose and plasma miR-221 and -222 levels in a cohort of patients with advanced CKD, which has not been explored in the past

Our results should be interpreted with caution, due to several inadequacies. The inability to conclude any causation due to the methodological flaws of a cohort study and small sample size limits generalizability. Dietary habits and concurrent medications may also induce bias. Considering this, we excluded patients who were on anabolic steroids at recruitment. We also analyzed certain classes of drugs, which may induce DM and obesity. All subjects also received standard dietary counseling by experienced dieticians at the time of dialysis training to ensure a similar dietary caloric intake. In addition, we also reduced the confounding effects of such by adding drug use, metabolic and nutritional parameters such as albumin, and lipid profile into the statistical analysis model. Adipose tissue samples may be contaminated with other kinds of cells during collection, such that the measured miRNA levels may not reflect the true adipose level. Similarly, the miRNA in adipose and plasma samples may be affected during specimen transport and processing, although studies have reported that miRNA is notably stable in archive samples [57], with no difference in levels detected for several years under appropriate storage conditions [58,59], and for 24 h at room temperature [60]. Nevertheless, we did not examine the serial change in plasma and adipose miRNA levels, levels of other associated biomarkers in the metabolic pathways such as Sirtuins, PPARs, interleukin (IL)-6, and TNF-alpha, and clinical outcomes beyond 2 years. Further studies to evaluate the associated biomarkers with a larger sample size and longer follow-up period should be advocated to clarify these inadequacies. Nonetheless, we evaluated insulin resistance by HOMA, rather than the gold standard tool of hyperinsulinemic euglycemic glucose clamp (HEGC) [61]. Hyperinsulinemic euglycemic glucose clamp is time-consuming, labor-intensive, tedious, and therefore impractical and seldom used in a real-life situation. Comparatively, HOMA is a commonly used alternative as it is less labor-intensive, more cost-effective, and practical [61]. Despite the above-listed limitations, our study remains the first in the literature that has quantified miR-221 and -222 in advanced CKD patients. We also illustrated their role in dialysis-associated obesity and glucose intolerance.

## 5. Conclusions

In conclusion, we demonstrated aberrant miRNA profiling in advanced CKD patients with DM. Our findings also showed adipose miR-221 and -222 predicted the progression of obesity, and insulin resistance after PD. We also indicated plasma miR-222 can accurately predict the onset of NODM after PD. Further studies are warranted to explore the use of miRNA profiling in risk stratification, and the potential benefits of miRNA-based therapy.

## Figures and Tables

**Figure 1 nutrients-14-04889-f001:**
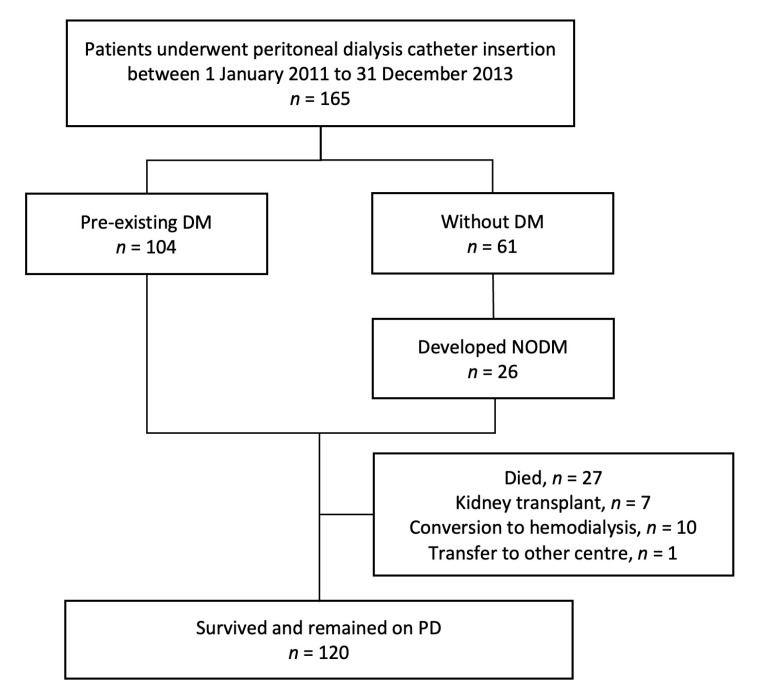
Flow chart.

**Figure 2 nutrients-14-04889-f002:**
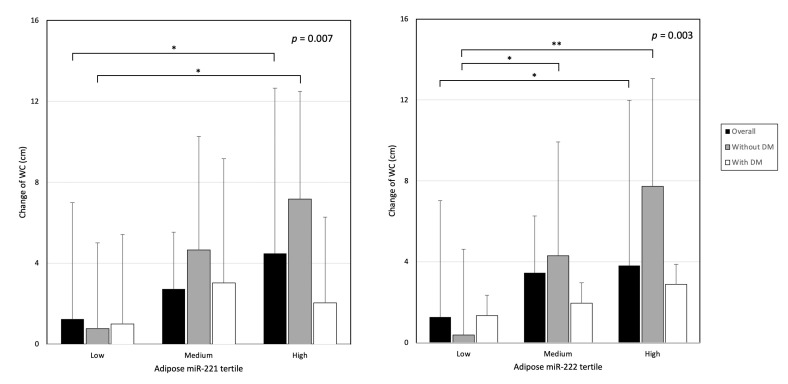
WC change according to adipose miR-221 and -222 tertiles. * denotes *p* < 0.05; ** denotes *p* < 0.01.

**Figure 3 nutrients-14-04889-f003:**
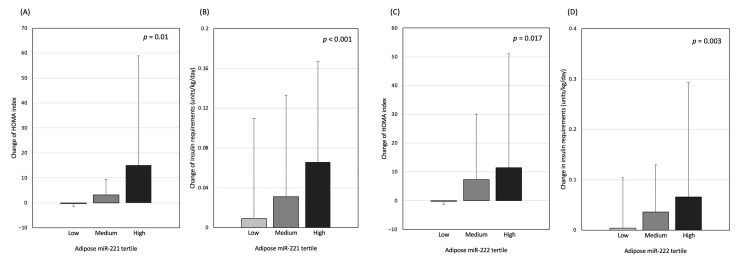
HOMA and daily insulin requirement change according to adipose (**A**,**B**) miR-221 and (**C**,**D**) -222 tertiles.

**Figure 4 nutrients-14-04889-f004:**
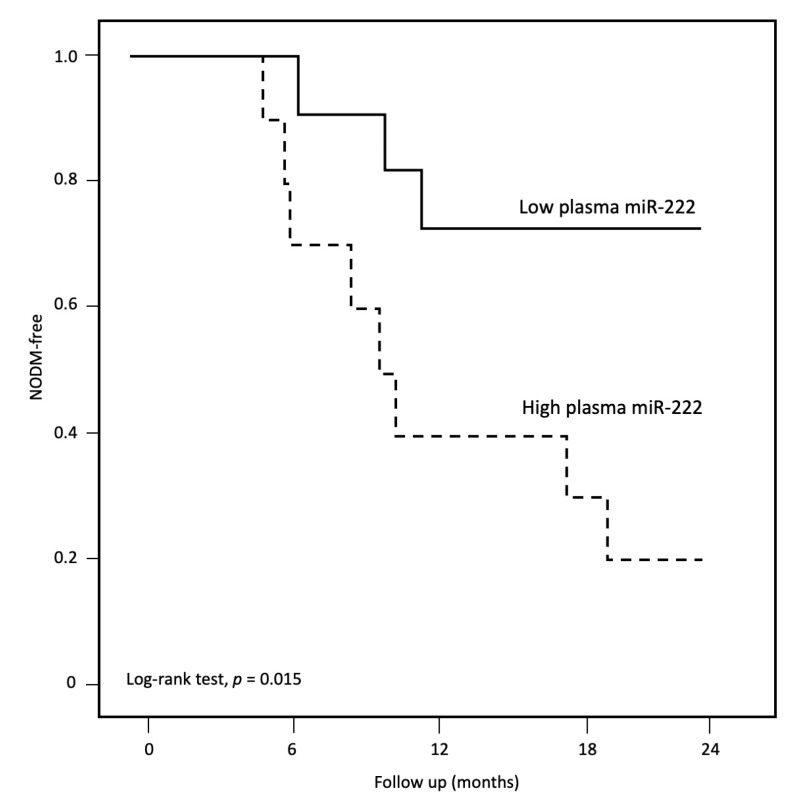
Kaplan–Meier curve of NODM in non-DM patients.

**Table 1 nutrients-14-04889-t001:** Clinical characteristics of recruited subjects.

	With DM (*n* = 104)	Without DM (*n* = 61)	*p*-Value
Age	60.9 ± 9.0	54.1 ± 12.9	*p* < 0.001 ^a^
Male sex, No. (%)	86 (82.7%)	40 (65.6%)	*p* = 0.01 ^b^
Primary renal disease, No. (%)			*p* < 0.001 ^b^
Diabetes mellitus	85 (81.7%)	0 (0%)	
Hypertension	3 (2.9%)	12 (19.7%)	
Glomerulonephritis	10 (9.6%)	27 (44.3%)	
Polycystic kidney disease	2 (1.9%)	1 (1.6%)	
Urological	0 (0%)	6 (9.8%)	
Others	1 (1.0%)	2 (3.3%)	
Unknown	3 (2.9%)	13 (21.3%)	
Comorbidities			
Ischemic heart disease	39 (37.5%)	7 (11.5%)	*p* < 0.001 ^b^
Cerebrovascular accident	23 (22.1%)	9 (14.8%)	*p* = 0.2 ^b^
Peripheral vascular disease	12 (11.5%)	0 (0%)	*p* = 0.006 ^b^
Charlson comorbidities index	7.3 ± 1.9	4.3 ± 2.0	*p* < 0.001 ^a^
Residual GFR (ml/min/1.73 m^2^)	4.4 ± 2.8	3.1 ± 2.3	*p* = 0.01 ^a^
nPCR (g/kg/day)	1.1 ± 0.3	1.1 ± 0.2	*p* = 0.2 ^a^
Laboratory parameters			
Urea (mmol/L)	30.4 ± 7.4	31.0 ± 8.0	*p* = 0.7 ^a^
Creatinine (umol/L)	811 ± 236	921 ± 303	*p* = 0.02 ^a^
Albumin (g/L)	34.9 ± 4.2	35.9 ± 4.7	*p* = 0.2 ^a^
High-sensitive C-reactive protein (mg/L)	10.7 ± 26.3	15.6 ± 33.5	*p* = 0.3 ^a^
Cholesterol, total (mmol/L)	4.5 ± 1.2	4.6 ± 1.2	*p* = 0.7 ^a^
High-density lipoprotein (mmol/L)	1.2 ± 0.4	1.3 ± 0.3	*p* = 0.1 ^a^
Low-density lipoprotein (mmol/L)	2.5 ± 1.0	2.7 ± 1.1	*p* = 0.5 ^a^
Triglycerides (mmol/L)	1.6 ± 0.9	1.3 ± 0.7	*p* = 0.03 ^a^
Peritoneal characteristics			
D/P4	0.69 ± 0.13	0.69 ± 0.12	*p* = 0.8 ^a^
MTAC	10.6 ± 4.9	11.3 ± 5.2	*p* = 0.4 ^a^
Pulse wave velocity			
Carotid-femoral (m/s)	12.0 ± 2.4	10.5 ± 1.9	*p* < 0.001 ^a^
Carotid-radial (m/s)	10.5 ± 1.3	10.6 ± 1.4	*p* = 0.6 ^a^
Medication use			
Statins	15 (24.6%)	63 (60.6%)	*p* < 0.001 ^b^
Beta blockers	44 (72.1%)	83 (79.8%)	*p* = 0.3 ^b^
Thiazide diuretics	19 (31.1%)	49 (47.1%)	*p* = 0.04 ^b^
Antidepressants	3 (4.9%)	6 (5.8%)	*p* = 0.8 ^b^
Antipsychotics	1 (1.6%)	4 (3.8%)	*p* = 0.4 ^b^
Antiepileptics	5 (8.2%)	1 (1.0%)	*p* = 0.02 ^b^

Data are presented as mean ± standard deviation and compared by paired Student’s *t*-test ^a^ and chi-square test ^b^. DM, diabetes mellitus; GFR, glomerular filtration rate; nPCR, normalized protein catabolic rate; D/P4, dialysate-to-plasma ratio of creatinine at 4 h; MTAC, mass transfer area coefficient of creatinine.

**Table 2 nutrients-14-04889-t002:** Anthropometry, glycemic, and miRNA profile at baseline.

	With DM (*n* = 104)	Without DM (*n* = 61)	*p*-Value
Anthropometry			
Body weight (kg)	68.3 ± 13.6	62.0 ± 14.9	*p* = 0.011
Body height (m)	164 ± 8	163 ± 9	*p* = 0.4
BMI (kg/m^2^)	25.3 ± 3.9	23.3 ± 4.4	*p* = 0.006
WC (cm)	91.8 ± 9.9	84.0 ± 12.5	*p* < 0.001
Glycemic profile			
HbA1c (%)	6.9 ± 1.1	5.5 ± 0.4	*p* < 0.001
Fasting glucose (mmol/L)	6.4 ± 2.2	4.9 ± 0.6	*p* < 0.001
Fasting insulin (mIU/L)	33.1 ± 44.3	13.5 ± 10.7	*p* = 0.002
C-peptide (ng/mL)	8.4 ± 5.7	10.3 ± 8.8	*p* = 0.1
HOMA	9.2 ± 12.4	2.9 ± 2.2	*p* < 0.001
microRNA level (fold)			
miR-221			
Adipose	1.13 ± 1.19	0.70 ± 0.85	*p* = 0.02
Plasma	2.52 ± 1.20	3.25 ± 1.42	*p* = 0.04
miR-222			
Adipose	1.90 ± 1.97	1.16 ± 1.29	*p* = 0.01
Plasma	0.13 ± 0.06	0.18 ± 0.16	*p* = 0.08

Data are presented as mean ± standard deviation and compared by paired Student’s *t*-test. BMI, body mass index; WC, waist circumference; WHR, waist-to-hip ratio; HbA1c, glycated haemoglobin A1c; HOMA, homeostatic model assessment.

**Table 3 nutrients-14-04889-t003:** Longitudinal change in (A) anthropometry, (B) glycaemic profile, and (C) insulin requirements.

(A) Anthropometry					
	Baseline	At 12th Month (Change)	*p*-Value	At 24th Month (Change)	*p*-Value
Body weight (kg)	65.9 ± 14.4	66.9 ± 13.4(1.8 ± 5.1)	*p* < 0.001	66.8 ± 13.7(0.5 ± 4.7)	*p* = 0.3
BMI (kg/m^2^)	24.5 ± 4.2	24.9 ± 3.9(0.7 ± 1.9)	*p* < 0.001	25.1 ± 4.0(0.2 ± 1.9)	*p* = 0.3
WC (cm)	88.8 ± 11.6	91.4 ± 11.6(2.9 ± 5.8)	*p* < 0.001	92.7 ± 12.5(1.3 ± 4.0)	*p* = 0.002
**(B) Glycaemic Profile**					
	**Baseline**	**At 12th Month** **(Change)**	***p*-Value**	**At 24th Month** **(Change)**	***p*-Value**
HbA1c (%)	6.3 ± 1.1	6.2 ± 1.1 (−0.2 ± 0.9)	*p* = 0.04	6.2 ± 1.2(+0.1 ± 0.8)	*p* = 0.4
Fasting glucose (mmol/L)	5.8 ± 1.9	6.5 ± 1.9 (+0.7 ± 2.0)	*p* < 0.001	6.0 ± 1.6(−0.3 ± 2.1)	*p* = 0.1
Fasting insulin (mIU/L)	25.6 ± 36.6	43.1 ± 114.9 (+19.1 ± 89.4)	*p* = 0.02	32.8 ± 73.8(−16.4 ± 83.0)	*p* = 0.07
C-peptide (ng/mL)	9.1 ± 7.1	9.3 ± 5.1 (+0.3 ± 8.2)	*p* = 0.02	8.0 ± 4.0(−1.2 ± 4.6)	*p* = 0.7
HOMA	6.8 ± 10.2	12.1 ± 32.5 (+ 6.3 ± 27.2)	*p* = 0.02	9.2 ± 22.4(−4.4 ± 20.2)	*p* = 0.04
**(C) Insulin Requirements**					
	**Total Daily Insulin Prescription** **(Units/Day)**		**Body Weight-Adjusted Daily Insulin Prescription** **(Units/kg/Day)**		
	**Dosage**	**Change**	**Dosage**	**Change**	***p*-Value**
Baseline	7.78 ± 14.10		0.11 ± 0.20		
At 6th month	11.05 ± 15.97	+2.72 ± 10.85	0.15 ± 0.23	+0.03 ± 0.15	*p* = 0.006
At 12th month	11.76 ± 18.07	+0.57 ± 7.75	0.15 ± 0.23	−0.005 ± 0.09	*p* = 0.5
At 18th month	11.61 ± 18.25	−0.22 ± 8.56	0.15 ± 0.24	+0.001 ± 0.12	*p* = 0.9
At 24th month	11.00 ± 18.33	−0.79 ± 10.27	0.13 ± 0.23	−0.017 ± 0.12	*p* = 0.1

Data are presented as mean ± standard deviation and compared by paired Student’s *t* test. BMI, body mass index; WC, waist circumference; HbA1c, glycated haemoglobin A1c; HOMA, homeostatic model assessment.

**Table 4 nutrients-14-04889-t004:** Linear regression model for predictors of changes in (A) anthropometry, and (B) glycemic profile.

(A) Change in Anthropometry						
	BW Change at 12th Month		BMI Change at 12th Month		WC Change at 12th Month	
	Beta (95% CI)	*p*-value	Beta (95% CI)	*p*-value	Beta (95% CI)	*p*-value
Adipose miR-221	1.44 (0.31–2.57)	*p* = 0.01	0.56 (0.15–0.98)	*p* = 0.009	1.82 (0.57–3.07)	*p* = 0.005
Adipose miR-222					1.35 (0.08–2.63)	*p* = 0.038
hsCRP			0.15 (0.00–0.03)	*p* = 0.048		
Baseline BMI			−0.12 (−0.22–−0.03)	*p* = 0.01		
Albumin	0.25 (0.02–0.47)	*p* = 0.03				
Pre-existing DM					−2.35 (−4.69–−0.01)	*p* = 0.049
Dialysate dextrose load					0.04 (0.01–0.07)	*p* = 0.003
Antidepressant use	−4.77 (−8.61–−0.94)	*p* = 0.015	−1.89 (−3.29–−0.46)	*p* = 0.01		
**(B) Change in Glycemic Profile**						
	**Plasma Insulin Change at 12th Month**		**HOMA Change at 12th Month**		**Insulin Requirement Change at 6th Month**	
	Beta (95% CI)	*p*-value	Beta (95% CI)	*p*-value	Beta (95% CI)	*p*-value
Adipose miR-221	28.61 (10.73–46.50)	*p* = 0.002	8.16 (2.80–13.53)	*p* = 0.003	0.05 (0.006–0.09)	*p* = 0.02
Adipose miR-222	21.61 (3.22–40.01)	*p* = 0.02	6.59 (1.13–12.05)	*p* = 0.018	0.06 (0.02–0.11)	*p* = 0.002
Baseline HOMA	5.91 (4.19–7.63)	*p* < 0.001	1.77 (1.26–2.28)	*p* < 0.001	0.008 (0.004–0.011)	*p* < 0.001
Pre-existing DM					0.19 (0.12–0.27)	*p* < 0.001

Covariates in the model: Age, pre-existing DM, DP4, daily dextrose load, baseline HOMA, PCR, BMI, WC, serum albumin, hsCRP, total cholesterol, triglycerides, low- and high-density lipoprotein, prescription of antidepressants. Covariates in the model: Age, pre-existing DM, hsCRP, DP4, daily dextrose load, baseline HOMA, nPCR, BMI, WC. BW, body weight; BMI, body mass index; WC, waist circumference; CI, confidence interval; DM, diabetes mellitus; hsCRP, high-sensitive C-reactive protein; DM, diabetes mellitus; HOMA, homeostatic model assessment; nPCR, normalized protein catabolic rate.

## Data Availability

The data generated in this study are available from the corresponding author on reasonable request, subject to approval from the local authority.

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
