# Peer review of "Adipose and Plasma microRNAs miR-221 and 222 Associate with Obesity, Insulin Resistance, and New Onset Diabetes after Peritoneal Dialysis"

_nutrients, 2022, doi:10.3390/nu14224889_

Round 1

Reviewer 1 Report

The authors are investigating miR-221 and miR-222 in PD patients. miR-221, 222 in PD patients have not been investigated before and are of interest. However, this manuscript raises some questions.

Plasma miR-222 levels were lower in the DM group (table 2). This contradicts previous reports that miR-222 is elevated in DM and obesity. The level of miR-222 is known to change with insulin and other factors, suggesting that it may be affected by medication. Alternatively, the method for detecting miR222 may not be appropriate. Therefore, the data may not be appropriate for other results as well.

In the DM group, there was no significant difference in the amount of change in WC between the group with high and low levels of adipose tissue miR221 and miR222 (Fig 2). Only the non-DM group has a significant difference. Could miR221 and 222 not be predictors of WC in patients with pre-existing DM? Although the authors did not specifically discuss about this, it is possible that the difference in diet and medication between the DM and non-DM groups may be reflected in these results. Since the contents of diet and medication are unknown, it seems inappropriate to examine whether baseline miR221 and 222 values ​​can predict obesity and glucose tolerance.

Among the non-DM groups in this study, body weight differs by more than 10% at baseline between the with NODM group and the without NODM group. Since the two groups were similar in height, it is thought that the with NODM was originally more obese. Therefore, it is natural that miR-222, which is high in obesity, is high in the with NODM onset group. The authors state that they performed a multivariate analysis, but do not state which variables were included in the analysis. Also, since the number of cases is only 60, it may not be possible to include a sufficient number of variables for multivariate analysis.

Author Response

Point 1: Plasma miR-222 levels were lower in the DM group (table 2). This contradicts previous reports that miR-222 is elevated in DM and obesity. The level of miR-222 is known to change with insulin and other factors, suggesting that it may be affected by medication. Alternatively, the method for detecting miR222 may not be appropriate. Therefore, the data may not be appropriate for other results as well.

Response 1: We followed the method of microRNA detection according to the manufacturer instruction and protocol. In short, we extracted the RNA and synthesized the cDNA, followed by the amplification by real-time polymerase chain reaction (RT-PCR) method and detection of microRNA by the Taqman miRNA assays. This traditional method has been used in various trials in the past [1, 2].

Line 461 – 478: We acknowledged that our cohort with DM had a numerically lower though insignificant plasma miR-222 (0.13 ± 0.06-fold vs 0.18 ± 0.16-fold, p = 0.08), which is in contrary to the existing literature in general population with type 2 diabetes and gestational diabetes [3, 4]. The paradoxical observations can be explained by the distinct have distinct biochemical characteristics in diabetic patients with advanced CKD. It has been reported that nearly one third of diabetic patients with CKD [5], and 17% of diabetic patients on PD [6] had normal HbA1c even without antidiabetic agent use. The decreased renal and hepatic insulin clearance, decline in renal gluconeogenesis, deficient catecholamine release, diminished food intake and defective energy utilization when renal function declines lead to the spontaneous improvement of glycemic control - a phenomenon called “burnt-out diabetes” [5, 6]. In CKD, there is also an accumulation of RNases that are responsible for the turnover and degradation of microRNA [7]. With that, past study has confirmed a different circulating microRNA profile in patients with CKD compared to healthy individuals [8]. Since miR-221 & 222 levels in CKD patients were not explored in the past, this represents a strength of our study as we are the first to characterized and quantify the miR-221 and 222 levels in cohort of patients with advanced CKD.

Reference:

  1. Arner E, Mejhert N, Kulyté A, Balwierz PJ, Pachkov M, Cormont M, Lorente-Cebrián S, Ehrlund A, Laurencikiene J, Hedén P, Dahlman-Wright K, Tanti JF, Hayashizaki Y, Rydén M, Dahlman I, van Nimwegen E, Daub CO, Arner P. Adipose tissue microRNAs as regulators of CCL2 production in human obesity. Diabetes. 2012 Aug;61(8):1986-93. doi: 10.2337/db11-1508. Epub 2012 Jun 11. PMID: 22688341; PMCID: PMC3402332.
  2. Scoditti E, Carpi S, Massaro M, Pellegrino M, Polini B, Carluccio MA, Wabitsch M, Verri T, Nieri P, De Caterina R. Hydroxytyrosol Modulates Adipocyte Gene and miRNA Expression Under Inflammatory Condition. Nutrients. 2019 Oct 17;11(10):2493. doi: 10.3390/nu11102493. PMID: 31627295; PMCID: PMC6836288.
  3. Sadeghzadeh S, Dehghani Ashkezari M, Seifati SM, Vahidi Mehrjardi MY, Dehghan Tezerjani M, Sadeghzadeh S, Ladan SAB. Circulating miR-15a and miR-222 as Potential Biomarkers of Type 2 Diabetes. Diabetes Metab Syndr Obes. 2020 Oct 2;13:3461-3469. doi: 10.2147/DMSO.S263883. PMID: 33061506; PMCID: PMC7537850.
  4. Filardi T, Catanzaro G, Grieco GE, Splendiani E, Trocchianesi S, Santangelo C, Brunelli R, Guarino E, Sebastiani G, Dotta F, Morano S, Ferretti E. Identification and Validation of miR-222-3p and miR-409-3p as Plasma Biomarkers in Gestational Diabetes Mellitus Sharing Validated Target Genes Involved in Metabolic Homeostasis. Int J Mol Sci. 2022 Apr 12;23(8):4276. doi: 10.3390/ijms23084276. PMID: 35457094; PMCID: PMC9028517.
  5. Kalantar-Zadeh K, Derose SF, Nicholas S, Benner D, Sharma K, Kovesdy CP. Burnt-out diabetes: impact of chronic kidney disease progression on the natural course of diabetes mellitus. J Ren Nutr. 2009 Jan;19(1):33-7. doi: 10.1053/j.jrn.2008.11.012. PMID: 19121768; PMCID: PMC2652655.
  6. Abe M, Hamano T, Hoshino J, Wada A, Nakai S, Masakane I. Rate of the "burnt-out diabetes" phenomenon in patients on peritoneal dialysis. Diabetes Res Clin Pract. 2018 Sep;143:254-262. doi: 10.1016/j.diabres.2018.07.026. Epub 2018 Jul 27. PMID: 30056189.
  7. Muralidharan J, Ramezani A, Hubal M, Knoblach S, Shrivastav S, Karandish S, Scott R, Maxwell N, Ozturk S, Beddhu S, Kopp JB, Raj DS. Extracellular microRNA signature in chronic kidney disease. Am J Physiol Renal Physiol. 2017 Jun 1;312(6):F982-F991. doi: 10.1152/ajprenal.00569.2016. Epub 2017 Jan 11. PMID: 28077372; PMCID: PMC5495885.
  8. Neal CS, Michael MZ, Pimlott LK, Yong TY, Li JY, Gleadle JM. Circulating microRNA expression is reduced in chronic kidney disease. Nephrol Dial Transplant. 2011 Nov;26(11):3794-802. doi: 10.1093/ndt/gfr485. Epub 2011 Sep 2. PMID: 21891774.

Point 2: In the DM group, there was no significant difference in the amount of change in WC between the group with high and low levels of adipose tissue miR221 and miR222 (Fig 2). Only the non-DM group has a significant difference. Could miR221 and 222 not be predictors of WC in patients with pre-existing DM? Although the authors did not specifically discuss about this, it is possible that the difference in diet and medication between the DM and non-DM groups may be reflected in these results. Since the contents of diet and medication are unknown, it seems inappropriate to examine whether baseline miR221 and 222 values ​​can predict obesity and glucose tolerance.
Response 2: Line 133 - 136: Although we did not objectively measure caloric and fat intake, the dietary intake, all recruited subjects received standard dietary counseling by experienced dietitians at the time of dialysis training. In general, they were advised to have energy intake 25 to 35 kcal/kg/day (depending on their body mass index [BMI]), 25–30% of which is fat, and protein 1.0–1.2 g/kg/day.

Line 129 – 131, Table 1: As for the effect from medications, we analyzed the use of drug class which commonly cause DM and obesity during their therapeutic use, namely statins, beta blockers, thiazide diuretics, antidepressants, antipsychotics, and antiepileptics. The results are added to the revised Table 1. In short, we observed lower prescription rates of statins and thiazide diuretics, and a higher rate of antiepileptics in patients with preexisting DM. In the univariate linear regression model, there is a significant association between statins and baseline BMI (unstandardized B 1.73, 95% confidence interval [CI] 0.35 – 3.11, p = 0.01), but not with BW (p = 0.2) and WC (p = 0.2). Statin use is not associated with 1-year changes in BW (p = 0.8), BMI (p = 0.7) and WC (p = 0.6). On the other hand, the prescription of antidepressants is associated with 1-year changes in BW (unstandardized B = -4.57; 95% CI -8.43 - -0.72, p = 0.02) and BMI (unstandardized B = -1.82, 95% CI -3.29 - -0.35, p = 0.02), but not WC (p = 0.5).

Table 4A: To address the confounding effects, we also newly add metabolic and nutritional parameters such as albumin, and lipids profile, and antidepressants use into the multivariate regression model. In essence, adipose miR-221 (unstandardized B 1.82, 95% CI 0.57 – 3.07, p = 0.005) and 222 (unstandardized B 1.35, 95% CI 0.08 – 2.63, p = 0.038) remain significantly associated with WC change in the first year, while adipose miR-221 is also associated with BW (unstandardized B 1.44, 95% CI 0.31 – 2.57, p = 0.01) and BMI (unstandardized B 0.56, 95% CI 0.15 – 0.98, p = 0.009) changes in the first year. Antidepressant use is also associated with BW (unstandardized B -4.77, 95% CI -8.61 – -0.94, p = 0.015) and BMI changes (unstandardized B -1.89, 95% CI -3.29 – -0.46, p = 0.01).

Point 3: Among the non-DM groups in this study, body weight differs by more than 10% at baseline between the with NODM group and the without NODM group. Since the two groups were similar in height, it is thought that the with NODM was originally more obese. Therefore, it is natural that miR-222, which is high in obesity, is high in the with NODM onset group. The authors state that they performed a multivariate analysis, but do not state which variables were included in the analysis. Also, since the number of cases is only 60, it may not be possible to include a sufficient number of variables for multivariate analysis.
Response 3: Line 158 – 160: Factors such as age, gender, baseline fasting glucose level, waist circumference, body mass index, dialysate glucose load, prescription of icodextrin containing dialysate, and serum high-sensitive C-reactive protein (hsCRP), which can potentially exert confounding effects and influence outcomes were added into the multivariate Cox regression analysis for new onset diabetes mellitus (NODM). The list of factors is added to the Statistical Analysis section.

Line 479 – 481, 496 – 498: We acknowledge the small sample size in the non-DM group at baseline may induce bias during the analyses. This limitation is highlighted in the revised manuscript. We also suggest further study with a larger sample size to address this issue.

Reviewer 2 Report

The paper entitled <Adipose and plasma microRNAs miR-221 & 222 associate with obesity, insulin resistance, and new-onset diabetes after peritoneal dialysis>. authors GCK Chan et al. talk about the adipose and plasma miR-221 18 & 222 levels and their association with adiposity and glucose intolerance in patients with PD.

The miR-221 18 & 222 levels are associated with glucose levels and insulin resistance. Previously, this kind of research was not conducted on kidney patients. It is wise that the blood sample was left when the patient was admitted to peritoneal dialysis treatment, but I am not sure that keeping the sample for such a long time does not affect the accuracy of the results.

Certainly, the methodology was carefully thought out, and the results reflected the goal of the work. The statistical processing of the data is adequate, as is the presentation of the obtained results.

The number of respondents is enviable, bearing in mind that a small number of patients are treated with peritoneal dialysis in any part of the world.

The work is an interesting contribution to a significant problem present in nephrology.

Author Response

Point 1: The miR-221 18 & 222 levels are associated with glucose levels and insulin resistance. Previously, this kind of research was not conducted on kidney patients. It is wise that the blood sample was left when the patient was admitted to peritoneal dialysis treatment, but I am not sure that keeping the sample for such a long time does not affect the accuracy of the results.

Response 1: Line 74 – 76: The blood sample was obtained on the day of peritoneal dialysis catheter insertion, which was then immediately processed, centrifuged, and stored at -4°C overnight for further extraction of RNA and microRNA.

Line 489 – 493: The stability of microRNA under different storage condition has been extensively studied in the past. The microRNA levels are notably stable even in archive samples [1], with no difference in levels detected for several years under appropriate storage condition [2, 3], and for 24 hours at room temperature [4].

The stability of microRNA under different storage condition has been extensively studied in the past. It was reported that the microRNA levels were stable for at least 24 hours at room temperature in whole blood [1]. Nevertheless, the microRNA does not degrade significantly even after several years with appropriate storage condition [2, 3].

Reference:

  1. Peiró-Chova L, Peña-Chilet M, López-Guerrero JA, García-Giménez JL, Alonso-Yuste E, Burgues O, Lluch A, Ferrer-Lozano J, Ribas G. High stability of microRNAs in tissue samples of compromised quality. Virchows Arch. 2013 Dec;463(6):765-74. doi: 10.1007/s00428-013-1485-2. Epub 2013 Oct 3. PMID: 24197449.
  2. Ge Q, Zhou Y, Lu J, Bai Y, Xie X, Lu Z. miRNA in plasma exosome is stable under different storage conditions. Molecules. 2014 Jan 27;19(2):1568-75. doi: 10.3390/molecules19021568. PMID: 24473213; PMCID: PMC6271968.
  3. Balzano F, Deiana M, Dei Giudici S, Oggiano A, Baralla A, Pasella S, Mannu A, Pescatori M, Porcu B, Fanciulli G, Zinellu A, Carru C, Deiana L. miRNA Stability in Frozen Plasma Samples. Molecules. 2015 Oct 20;20(10):19030-40. doi: 10.3390/molecules201019030. PMID: 26492230; PMCID: PMC6331950.
  4. Glinge C, Clauss S, Boddum K, Jabbari R, Jabbari J, Risgaard B, Tomsits P, Hildebrand B, Kääb S, Wakili R, Jespersen T, Tfelt-Hansen J. Stability of Circulating Blood-Based MicroRNAs - Pre-Analytic Methodological Considerations. PLoS One. 2017 Feb 2;12(2):e0167969. doi: 10.1371/journal.pone.0167969. PMID: 28151938; PMCID: PMC5289450.

Point 2: The number of respondents is enviable, bearing in mind that a small number of patients are treated with peritoneal dialysis in any part of the world.
Response 2: Line 392 – 397: We acknowledge that a smaller number of patients were treated with PD, compared to hemodialysis in many countries. However, the global uptake of PD has increased over the past decade in many parts of the world, including USA [1]. Compared to haemodialysis, PD incurs a lower cost and less demand on infrastructure, staff expertise, and patients training [2, 3]. As a home dialysis therapy, it also minimizes frequent hospital visits, which are particularly advantageous amid the novel coronavirus-2019 (COVID-19) pandemic. With that, the burden of metabolic complications after PD such as excessive weight gain and adiposity will become a global challenge to tackle among dialysis population. This issue is added to the Discussion section.

Reference:

  1. Li PK, Chan GC, Chen J, Chen HC, Cheng YL, Fan SL, He JC, Hu W, Lim WH, Pei Y, Teo BW, Zhang P, Yu X, Liu ZH. Tackling Dialysis Burden around the World: A Global Challenge. Kidney Dis (Basel). 2021 May;7(3):167-175. doi: 10.1159/000515541. Epub 2021 Apr 29. PMID: 34179112; PMCID: PMC8215964.
  2. Li PK, Chow KM, Van de Luijtgaarden MW, Johnson DW, Jager KJ, Mehrotra R, Naicker S, Pecoits-Filho R, Yu XQ, Lameire N. Changes in the worldwide epidemiology of peritoneal dialysis. Nat Rev Nephrol. 2017 Feb;13(2):90-103. doi: 10.1038/nrneph.2016.181. Epub 2016 Dec 28. PMID: 28029154.
  3. Kaplan JM, Niu J, Ho V, Winkelmayer WC, Erickson KF. A Comparison of US Medicare Expenditures for Hemodialysis and Peritoneal Dialysis. J Am Soc Nephrol. 2022 Nov;33(11):2059-2070. doi: 10.1681/ASN.2022020221. Epub 2022 Aug 18. PMID: 35981764.

Reviewer 3 Report

In this observational cohort study, Chan et al. demonstrated that both serum and adipose tissue miR-221 & 222 levels are higher in diabetic incident PD patients than those in non-DM incident PD patients. Since previous study showed that miR-221 & 222 regulate adipocyte differentiation, they hypothesized that these levels could affect the adipose tissue phenotype in PD patients. By clustering the incident PD patients according to the miR-221 & 222 tertiles, they have shown the basal miR221 and 222 expressions in the adipose tissue are correlated with the future increase in waist circumference that represents the abdominal adipose tissue mass one year after the induction of PD. They are also associated with future HOMA and daily insulin requirements change one year after the PD initiation. Finally, they have shown that the incidence of NODM is higher in those with higher basal plasma levels of miR-222. They concluded that miR-221 & 222 associate with progression of obesity, insulin resistance and NODM after PD. Their study has shown the proof of concept of miR-221 and miR-222 have important roles in glucose metabolism in PD patients and provided the evidence that these microRNAs can be potential therapeutic target. The study is of clinical importance and have directly demonstrated the significance of these miR by using the human adipose tissue samples, However, several concerns ought to be addressed to confirm their hypothesis and clinical relevance of these microRNAs.

1.      Although they have shown that the serum levels of these miRNA can predict the incidence of NODM in the future, the correlation between WC or HOMA and ‘serum’ concentrations of miRNA 221 or 222 was not shown. Moreover, the correlation between serum and adipose tissue concentration of these microRNAs was not shown. These data can confirm the role of these microRNAs in the pathogenesis of insulin resistance in incident PD patients,

2.      The increase in glucose metabolism parameters and anthropometric parameters continued until one year while these parameters have become stable in the next year. However, the incidence of NODM increases until two months. They did not survey the incident at the third year. It is supposed that the number be stable in the future. In this sense, the long-term clinical changes cannot be predictable by these miRNA levels. It is considered that the clinical value of these parameters is limited.

3.      The expression levels of target genes including siurtuins or PPARs in the adipose tissues are better to be measured.

4.      They considered that adipose tissue changes after the induction of PD affects the insulin resistance or glucose metabolism in PD and these micoRNAs in adipocytes would regulate the differentiation or maturation of adipocytes. The author had better show the changes in serum levels of adiponectin, IL-6, or TNFa in order to show the connection of the adipocyte phenotype and systemic insulin resistance.

Author Response

Point 1: Although they have shown that the serum levels of these miRNA can predict the incidence of NODM in the future, the correlation between WC or HOMA and ‘serum’ concentrations of miRNA 221 or 222 was not shown. Moreover, the correlation between serum and adipose tissue concentration of these microRNAs was not shown. These data can confirm the role of these microRNAs in the pathogenesis of insulin resistance in incident PD patients.

Response 1: Line 171 – 172: We performed the correlation analyses as suggested. There was insignificant correlation between adipose miR-221 and plasma miR-221 (p = 0.7) and miR-222 (p = 0.2). Similarly, there was also a lack of correlation between adipose miR-222 and plasma miR-221 (p = 0.8) and miR-222 (p = 0.3).

Line 452 – 461: The absence of correlation between adipose and serum adipokine expression has been reported in the past [1]. As mentioned in the Discussion section, this finding reflects that adipose tissue only contributes to part of the circulating miRNAs since miRNAs are also produced in organs such as the brain, heart, liver, skeletal muscle, spleen, etc. While these miRNAs are preferably produced in white adipose tissue, specific membrane transporter is also required for their diffusion into the bloodstream. Therefore, plasma miRNA level reflects the endocrine action with the cumulated systemic effect.

In addition, our Supplementary Table S1 shows a significant correlation between plasma miR-222 and HOMA (r = -0.33, p = 0.015) but not with WC (r = 0.04, p = 0.8). This again reinforces role of miR-222 in the pathogenesis of insulin resistance in incident PD patients.

Reference:

1. Chan GC, Than WH, Kwan BC, Lai KB, Chan RC, Teoh JY, Ng JK, Chow KM, Fung WW, Cheng PM, Law MC, Leung CB, Li PK, Szeto CC. Adipose and serum zinc alpha-2-glycoprotein (ZAG) expressions predict longitudinal change of adiposity, wasting and predict survival in dialysis patients. Sci Rep. 2022 May 31;12(1):9087. doi: 10.1038/s41598-022-13149-6. PMID: 35641588; PMCID: PMC9158927.

Point 2: The increase in glucose metabolism parameters and anthropometric parameters continued until one year while these parameters have become stable in the next year. However, the incidence of NODM increases until two months. They did not survey the incident at the third year. It is supposed that the number be stable in the future. In this sense, the long-term clinical changes cannot be predictable by these miRNA levels. It is considered that the clinical value of these parameters is limited.

Response 2: Line 496 – 498: We agree that extension of follow up period permits a more comprehensive evaluation of the association between miRNA with long-term clinical changes. This limitation is addressed in the Discussion section. We suggest further study to evaluate the association between microRNA with incidence of long-term NODM and metabolic changes.

Point 3 & 4: The expression levels of target genes including siurtuins or PPARs in the adipose tissues are better to be measured. They considered that adipose tissue changes after the induction of PD affects the insulin resistance or glucose metabolism in PD and these microRNAs in adipocytes would regulate the differentiation or maturation of adipocytes. The author had better show the changes in serum levels of adiponectin, IL-6, or TNFa in order to show the connection of the adipocyte phenotype and systemic insulin resistance.

Response 3 & 4: Line 493 – 498: We did not evaluate other associated up- and down-stream parameters, and the biomarkers associated with the metabolic changes, such as Sirtuins, PPARs and IL-6 and TNF-alpha. Therefore, we are unable to infer the overall pathogenic pathway and establish the pathophysiological mechanisms with the clinical outcomes we studied. Further studies that study the simultaneous expression of Sirtuins, PPARs and IL-6 and TNF-alpha are warranted. This limitation and suggestion are highlighted in the revised Discussion section.

Round 2

Reviewer 1 Report

Even in the revised manuscript this time, the content of meals and insulin doses in the target patients are unknown. As the authors say, it is possible that mRNA in CKD patients is atypical. Even more so, the authors should design studies that can eliminate the effects of multiple confounding factors. There is insufficient information in this manuscript to support an unusual result.

Reviewer 3 Report

In spite of the limitation about the lack of long-term effects or of the association between serum and adipose tissue microRNA levels, the survival cure of NODM is of clinical relevance.